# Shikonin Derivatives Inhibit Inflammation Processes and Modulate MAPK Signaling in Human Healthy and Osteoarthritis Chondrocytes

**DOI:** 10.3390/ijms23063396

**Published:** 2022-03-21

**Authors:** Birgit Lohberger, Heike Kaltenegger, Nicole Eck, Dietmar Glänzer, Patrick Sadoghi, Andreas Leithner, Rudolf Bauer, Nadine Kretschmer, Bibiane Steinecker-Frohnwieser

**Affiliations:** 1Department of Orthopedics and Trauma, Medical University of Graz, Auenbruggerplatz 5, 8036 Graz, Austria; birgit.lohberger@medunigraz.at (B.L.); heike.kaltenegger@medunigraz.at (H.K.); nicole.eck@medunigraz.at (N.E.); dietmar.glaenzer@medunigraz.at (D.G.); patrick.sadoghi@medunigraz.at (P.S.); andreas.leithner@medunigraz.at (A.L.); 2Department of Pharmacognosy, Institute of Pharmaceutical Sciences, University of Graz, Beethovenstr. 8, 8010 Graz, Austria; rudolf.bauer@uni-graz.at (R.B.); nadine.kretschmer@uni-graz.at (N.K.); 3Ludwig Boltzmann Institute for Arthritis and Rehabilitation, Thorerstraße 26, 5760 Saalfelden, Austria

**Keywords:** osteoarthritis, shikonin, acetylshikonin, cyclopropylshikonin, MAPK signaling, STAT3, inflammation

## Abstract

Osteoarthritis (OA) is the most common joint disorder and is characterized by the degeneration of articular cartilage. To develop new therapeutic approaches, we investigated the effect of shikonin derivatives on inflammation, MMP expression, and the regulation of MAPK signaling in human healthy (HC) and OA chondrocytes (pCH-OA). Viability was analyzed using the CellTiter-Glo^®^ Assay. Inflammatory processes were investigated using a proteome profiler™ assay. Furthermore, we analyzed the effects of the shikonin derivatives by protein expression analysis of the phosphorylation pattern and the corresponding downstream gene regulation using RT-qPCR. Both HC and pCH-OA showed a dose-dependent decrease in viability after treatment. The strongest effects were found for shikonin with IC_50_ values of 1.2 ± 0.1 µM. Shikonin counteracts the inflammatory response by massively reducing the expression of the pro-inflammatory mediators. The phosphorylation level of ERK changed slightly. pJNK and pp38 showed a significant increase, and the downstream targets c/EBPs and MEF2c may play a role in the cartilage homeostasis. STAT3 phosphorylation decreased significantly and has a chondroprotective function through the regulation of cyclin D1 and Sox9. Our results demonstrate for the first time that shikonin derivatives have extensive effects on the inflammatory processes, MAPKs, and IL6/STAT3 downstream regulation in healthy and OA chondrocytes.

## 1. Introduction

Osteoarthritis (OA) is one of the most common joint disorders, leading to functional disability, especially in people aged over 50 years, causing loss to the economy and affecting social development [1]. OA is a disease that affects the entire joint, with pathological changes in all tissues, including articular cartilage degradation, subchondral bone thickening, osteophyte formation, synovitis, and degeneration of the ligaments and menisci. Numerous studies in recent years have shown that the development and expression of OA is causally influenced by several factors, including age, gender, and familial susceptibility, as well as local biomechanics, cartilage cell apoptosis, and the action of degenerative enzymes [2]. As OA develops, normally quiescent chondrocytes undergo a phenotypic change and become ‘activated’ cells, characterized by cell proliferation, cluster formation, and the increased production of both matrix proteins and matrix-degrading enzymes [3].

The involvement of inflammation, which leads to symptoms such as joint pain, swelling, and stiffness, is well studied [4]. In OA, chondrocytes and synovial fibroblasts become activated by mechanical stress, inflammatory cytokines, or altered amounts of the collagenolytic matrix metalloproteinases (MMPs) -1, -3, - and -13 [5]. The final stage of inflammatory joint diseases is characterized by excessive extracellular matrix (ECM) catabolism. Cartilage degradation in the course of inflammatory joint diseases is caused by chondrocytes that are stimulated to increase the release of MMPs, which after activation cause proteolysis of tendons, bone, and cartilage [6]. According to previous findings, phosphatidylinositol 3-kinase (PI3K) and its downstream serine/threonine kinase AKT are essential for the normal metabolism of joint tissue, are involved in the development of OA, and are correlated with OA pathogenesis [7]. The mitogen-activated protein kinase (MAPK) family consists of three members: the extracellular-signal regulated kinase (ERK), the c-Jun-N-terminal kinase (JNK), and the p38 kinase. They transduce signals from the cell membrane to the nucleus in response to a wide range of stimuli. Because the MAPKs are central regulators of additional cell signaling pathways that control cell proliferation, survival, MMP synthesis, and production of pain mediators, they have been considered potential therapeutic targets for several diseases, including OA [8].

Despite the considerable symptoms, the majority of patients with OA do not receive appropriate therapy. In most cases, treatment is palliative and reactive rather than a proactive and preventive intervention [2]. Current therapies for OA are limited to symptom-relieving drugs and total knee arthroplasty for severe cases. Drugs addressing the underlying biological causes of OA are not currently available [9]. Given the increasing individual and societal burden of OA, the treatment approach should evolve towards individualized therapy, which focuses on the specific needs of the patient. Research into new substances is an important step in this context.

Naturally occurring naphthoquinone derivatives, such as shikonin and derivatives thereof, can be found in different species of the Boraginaceae family, for example in the roots of *Lithospermum erythrorhizon* Siebold et Zucc or *Onosma paniculata* Bur. et Franch. These kinds of roots are traditionally used in Chinese medicine to treat, for example, infections, inflammatory diseases, and hemorrhagic diseases. The most studied derivative, shikonin, has also been shown to possess strong anti-cancer activity in several types of cancer cells [10,11,12,13,14]. The mechanisms of action included, for example, inhibition of cell proliferation, induction of apoptosis, and reduced cell migration and invasion through a variety of molecular signal transduction pathways. Another promising, naturally occurring shikonin derivative is acetylshikonin, which has also been reported to possess several pharmacological activities [15,16,17]. Furthermore, several attempts have been made to optimize the naturally occurring shikonin derivatives by modulating the structure. One of these novel synthetic derivatives is cyclopropylshikonin, which has already shown promising anti-cancer effects in melanoma cells [18].

Although there are a small number of publications related to OA in animal experiments, in the field of human cartilage research there are no findings, yet. Shikonin inhibited inflammatory responses in rabbit chondrocytes and showed chondroprotection in OA rabbit knee [19]. Furthermore, shikonin was shown to inhibit inflammatory processes and chondrocyte apoptosis by regulating the PI3K/AKT pathway in a rat model of OA [20,21,22]. The corresponding regulatory mechanisms in human OA chondrocytes and their influence by shikonin have not yet been investigated. The regulation of MAPKs and STAT3 by shikonin has mainly been shown so far in different tumor cell types, whereas nothing is known in the context of OA chondrocytes [23,24].

The aim of this study is to investigate the potential therapeutic application of shikonin derivatives in OA and their cell biological effectiveness. Therefore, we investigated the effect of shikonin, and its derivatives acetylshikonin and cyclopropylshikonin, on inflammation, MMP expression, and the regulation of MAPK signaling pathways. As inflammatory processes play a particularly important role in OA, we aim to gain more insight into the IL1-induced inflammatory pathways in chondrocytes and to investigate OA-specific differences in cell function. To obtain a representative cross-section of patient-specific differences, we use human primary chondrocytes from patients who have undergone a total knee arthroplasty.

## 2. Results and Discussion

### 2.1. Effects on Viability of Healthy Chondrocytes and Primary OA Chondrocytes

To study the effects of shikonin and its derivatives (Figure 1A) in healthy chondrocytes (HC) and primary OA chondrocytes (pCH-OA), cells were treated with various concentrations of the compounds of interest, and the dose-response relationship was analyzed. Shikonin and acetylshikonin are naturally occurring shikonin derivatives [15,16,17]. Cyclopropylshikonin was synthesized in a previous study and found to be the most cytotoxic compound out of a screening of novel shikonin derivatives [18]. It could be previously shown that the naphthoquinone scaffold, including the hydroxyl groups, is important for the activity of shikonin derivatives. Moreover, the side chain was reported to further influence the overall activity [18]. Both HC and pCH-OA showed a dose-dependent inhibition of cell viability after treatment with shikonin derivatives (Figure 1B). The strongest effects reflected by the lowest IC_50_ values were found for shikonin (IC_50_ 1.2 µM for HC; 1.3 µM for pCH-OA), indicating that the additional hydroxy group of the side chain mediates more cytototic effects than the esters of the other two compounds. The IC_50_ values of acetylshikonin were 2.4 µM and 2.1 µM, and the values for the novel derivate cyclopropylshikonin were 1.6 µM and 1.8 µM. The values are lower than those determined for melanoma cells and human embryonic kidney (HEK293) cells, indicating that human chondrocytes might be more sensitive to shikonin [18].

### 2.2. Inhibition of Inflammatory Cytokines and MMP-1 and -3

Research data in the past have shown that shikonin exerts anti-inflammatory effects in acute injuries in various animal models [22,25]. To investigate the possible suppression of the inflammatory process in human chondrocytes, we performed a proteome profile screening assay (Figure 2A). Only minor effects on HC cells by shikonin have been detected. Within the supernatants of HC cells treated with shikonin, the macrophage migration inhibitory factor (MIF), SerpineE1, and interleukin (IL)-8 levels were increased, while the C-X-C motif chemokine ligand (CXCL)12 was slightly reduced by the treatment. For MIF and SerpineE1, similar regulatory effects by shikonin, although more pronounced, have been found in pCH-OA cells. Interestingly, CXCL12 was regulated oppositely, showing a prominent increase by shikonin in pCH-OA cells. Secreted pro-inflammatory cytokines/chemokines C-C motif chemokine ligand (CCL)-1, -2, and -5, as well as the intercellular adhesion molecule (ICAM)-1, were heavily down regulated (Figure 2B). These data allow a first conclusion that shikonin counteracts the inflammatory response caused by IL-1β by massively reducing the expression of the pro-inflammatory mediators. To verify these data, gene expression analyses of IL6, matrix metalloproteinase (MMP)1, and MMP3, which are of particular importance in OA, were performed (Figure 2C). MMPs degrade the extracellular matrix and have a destructive potential in OA. Therefore, compounds that reduce their expression could have significant beneficial clinical effects. MMP1 belongs to the collagenases degrading interstitial collagens and MMP3 is a stromelysin that degrades non-collagen proteins. Both are induced by TNF-α and IL-1β [6]. Acetylshikonin and shikonin showed a highly significant reduction in these markers in pCH-OA cells, normally being upregulated in the clinical pattern of OA. On the other hand, treatment of the cells with cyclopropylshikonin revealed only minor effects. This inhibition of inflammation makes this group of substances particularly interesting for OA research.

### 2.3. Activation of AKT and ERK/JNK/p38 Phosphorylation

AKT is a serine/threonine kinase and plays a critical role in cell growth, differentiation, and survival [26]. Protein expression analysis showed a dose-dependent, significant increase in the phosphorylation status of AKT (pAKT) in HC after the first hour of treatment with all derivatives. pCH-OA showed the same effects at a lower intensity (Figure 3A). The corresponding densiometry of the pAKT/AKT ratio is shown in Figure 3B. Fu et al. [22] reported a downregulation of the PI3K/AKT pathway in a rat OA model. However, for this study, samples of arthritic tissue from OA rats were used, which were taken four days after a shikonin treatment.

Mitogen-activated protein kinase (MAPK) pathways have been shown to be activated in OA cartilage, and there is evidence, at least for ERK, that they can play a key role in the cartilage destruction seen in OA [8]. The levels of activated (phosphorylated) c-Jun N-terminal kinase (JNK) in human OA cartilage appear to be greater than the levels present in normal cartilage [27,28]. The phosphorylation of p38 was also higher in human OA compared to normal tissue, while phosphorylated ERK was found in both [28]. In our cellular system, the phosphorylation level of ERK changed only slightly; pJNK and pp38 showed a significant increase after treatment with the shikonin derivatives (Figure 4A), an effect that was observed in both HC and inflammatory pCH-OA cells. The corresponding densiometries of the pERK/ERK, pJNK/JNK, and pp38/p38 ratio are shown in Figure 4B. This type of regulation by shikonin has been described previously for human NB4 leukemia cells, where it resulted in increased phosphorylation of JNK and p38, while inhibiting ERK phosphorylation [29]. Gene expression analysis of the downstream targets RUNX family transcription factor 2 (RUNX2), CCAAT enhancer binding protein β (c/EBPβ), and myocyte enhancer factor 2c (MEF2c) were presented in Figure 4C. Several transcription factors have been implicated in the onset and progression of OA, including RUNX2 and c/EBPβ [30]. While the expression of RUNX2 decreases slightly due to the shikonin derivatives treatment, the expression of its most potent transcriptional partner, c/EBPβ, increased with high significance by shikonin and acetylshikonin (Figure 4C). As described by the literature, c/EBPβ overexpression might negatively affect the metabolism of the cartilage cells leading to the degradation of the cartilage matrix. On the other hand, c/EBPβ has been shown to play an important role in cartilage cell differentiation by modulating Sox9 expression. In addition, the c/EBP transcription factor family drives the differentiation of a number of cell types and plays a key role in regulating cell proliferation through interaction with cell cycle proteins [31]. Thus, an increase in the expression of cEBP beta in response to shikonin treatment in OA cells could indicate significant changes at the transcriptional level. Similarly, it is generally accepted that c/EBPβ acts as an important regulator of IL-6 signaling and plays an important role in transcriptional regulation of the IL-6 gene. From our observations, we further infer a possible relationship between IL-6 and cEBP. An increase in cEBP in OA cells could be accompanied by a decrease in IL-6 expression, suggesting that the upregulation of c/EBPβ by shikonin causes downregulation of IL-6 [32]. Further expression studies revealed MEF2c to be significantly overexpressed after treatment with the shikonin derivatives (Figure 4C).

### 2.4. Inhibition of STAT3 Phosphorylation and Influence on STAT 3 Downstream Targets

The IL6/signal transducer and activator of transcription 3 (STAT3) signaling plays an important role in OA pathogenesis and is considered a key transcription factor in promoting OA development [33]. IL6 binds first to its specific receptor and then to a common subunit (gp130), which triggers the STAT and ERK pathways. The formation of STAT3 dimers after phosphorylation is the major mechanism by which STAT3 regulates downstream gene expression and cellular biological processes [34].

Our data revealed that both in HC and pCH-OA the phosphorylation level of STAT3 significantly decreased in a dose-dependent manner after treatment with the shikonin derivatives (Figure 5A). The corresponding densiometries of the pSTAT3 ratio are shown in Figure 5B. The STAT3 blockade has a chondroprotective function, as STAT3 inhibition is known to decrease the expression of MMP3, MMP13, ADAMTS4, and ADAMTS5 in a mouse model [35].

In addition, we analyzed the gene expression of the STAT3 downstream targets cyclin D1, cMyc, suppressor of cytokine signaling 3 (SOCS3), and SRY-Box transcription factor 9 (Sox9). The relative gene expression values were presented in Figure 5C. After treatment with acetylshikonin and shikonin, both HC and pCH-OA revealed a highly significant reduction in cyclin D1 expression. Cyclopropylshikonin showed no changes. Cyclin D1 is a cell cycle regulator and promotes progression through the G_1_-S phase. In human lung cancer and pancreatic cells, shikonin suppressed cell proliferation through modulating the expression of the cell cycle regulators such as cyclin D1 or cMyc [36,37].

SOCS3 is a negative regulator of the JAK/STAT signaling pathway and was significantly increased by the shikonin derivatives treatment. In agreement with the protein expression data, this regulation is particularly well observed in HC. While STAT3 functions as a key modulator of Sox9 expression in nascent cartilage and developing chondrocytes, Sox9 itself is essential for chondrogenesis, promotes chondrocyte survival, and transcriptionally activates genes for many cartilage-specific structural components and regulatory factors [38]. Decreased STAT3 phosphorylation by treatment with the shikonin derivatives regulated Sox9 expression as it physically interacts with the promoter in response to stimulation (Figure 5C).

### 2.5. Effects on Apoptotic Induction by Investigating Survivin, Noxa, and Cleaved PARP

Survivin, a member of the inhibitors of apoptosis family, inhibits caspase activation and leads to negative regulation of apoptosis [39]. It has been shown that survivin is re-expressed in OA and that a high expression of survivin is correlated to a poor prognosis of patients. Therefore, survivin could represent a promising clinical target [40]. A highly significant reduction in survivin protein expression was observed in both HC and pCH-OA after treatment with 1.5 µM of the shikonin derivatives (Figure 6A). The corresponding densiometry is shown in Figure 6B. β-actin was used as a loading control. As the concentrations largely agree with the calculated IC_50_ values, the highly significant downregulation of survivin by elevated concentrations of shikonin derivatives might be a potential mechanism to induce apoptosis. Through these high concentrations, the cleavage of PARP (poly ADP ribose polymerase) could also be observed. During the apoptosis, caspase-3 and -7 are activated as executioner caspases and initiate PARP cleavage, whereby the cleavage of PARP is considered a hallmark of apoptosis. In healthy cells, PARP is responsible for DNA repair and genomic stability. PARP cleavage leads to its inactivation and, finally, to cell death. After treatment with shikonin and the derivatives, we found PARP cleavage for all of them. Thereby, the effects were stronger in pCH-OA cells than in HC cells, indicating that pCH-OA cells react more sensitively to apoptosis induction. In addition, we could show in previous experiments that shikonin derivatives can lead to Noxa upregulation [41]. Noxa is a pro-apoptotic BH-3-only protein and monitors the release of cytochrom c out of the mitochondria. It also activates down-stream caspases, and low levels of Noxa are associated with the insufficient induction of apoptosis. The associated Noxa upregulation has also been observed in melanoma cells [41]. In both HC and pCH-OA, increased shikonin derivatives concentrations caused such an increase in Noxa expression.

## 3. Materials and Methods

### 3.1. Origin of Shikonin Derivatives

Shikonin was purchased from Sigma Aldrich (St. Louis, MI, USA). Acetylshikonin was isolated from the dried roots of *Onosma paniculata*, as described previously [13]. In brief, freshly ground roots were extracted with petroleum ether by Soxhlet extraction. The extract was then subjected to a preparative Merck Hitachi HPLC system, consisting of an L-7100 pump, L-7200 autosampler, L-7455 diode array detector, and a D-7000 interface. Acetylshikonin was then isolated with the following column and method: VDSphere 100 RP-18 column, gradient and mobile phases: water (A) and ACN (B); 0–45 min: 70–100% B, 45–60 min: 100% B. (R)-1-(1,4-Dihydro-5,8-dihydroxy-1,4-dioxonaphthalen-2-yl)-4-methylpent-3-enyl 2-cyclopropyl-2-oxoacetate (“cyclopropylshikonin”) was prepared from shikonin as a starting material, as described in Kretschmer et al., 2021. In brief, acylation of shikonin was accomplished by Steglich esterification in dichloromethane with 2-cyclopropyl-2-oxoacetic and dicyclohexylcarbodiimide as a coupling reagent, as well as 4-dimethylaminopyridine as a catalyst. The purification and NMR data can be found in Kretschmer et al., 2021 [16]. The purity of all compounds was measured by HPLC and/or NMR and always exceeded 95%.

### 3.2. Cartilage Samples and Cell Culture

This study was carried out at the Department of Orthopedics and Trauma, Medical University of Graz, Austria, in accordance with the ethical standards of the responsible committee (Ethics committee of the Medical University of Graz, Austria, 31-133ex18/19) and the Helsinki Declaration. Patients (N = 15) aged 73.4 ± 5.8 years who underwent end-stage knee arthroplasty for knee OA were included and gave their written informed consent. Femoral cartilage was collected intraoperatively and digested using 2 mg/mL collagenase B (Gibco Invitrogen, Carlsbad, CA, USA) in a chondrocytes growth medium at 37 °C for 24 h. After filtration using a 40 µm filter and centrifugation, the primary cells were seeded in cell culture flasks. Cell passages 1 and 2 were used for experiments. Pre-stimulation with 10 ng/mL IL-1β were performed 1 h before the respective experiments. Human healthy chondrocytes (HC), isolated from a 65-year-old Caucasian man, purchased from Cell Application, Inc., (San Diego, CA, USA) were used as healthy comparison cells. The chondrocyte growth medium consists of DMEM/F12, supplemented with 10% fetal bovine serum (FBS), 1% Penicillin-Streptomycin (5.000 U/mL), 1% L-Glutamine, 1% Insulin-Transferrin-Selen, 0.01% TGF-β (10 ng/mL), and 0.01% FGF (10 ng/mL; all Gibco Invitrogen). Cells were kept in a humidified 5% CO_2_ atmosphere at 37 °C.

The cells were seeded in appropriate cell culture vessels and treated with 0 µM (control), 0.5 µM, and 1.5 µM shikonin derivatives after a growth period overnight. For the viability analysis, the protein isolation of the detection of survivin, Noxa, and cleaved PARP and the RNA isolation, the incubation period was 24 h. As the phosphorylation process is very fast, the proteins for the analyses of the AKT and MAPK pathways were already isolated 1 h after treatment.

### 3.3. Treatment Procedure

Shikonin (MW 288.29 g/mol), acetylshikonin (330.33 g/mol), and cyclopropylshikonin (MW 384.38 g/mol) were dissolved in ethanol in a stock solution of 10 mM. The further dilution step to the stock solution II with the concentration of 100 µM was performed in the respective culture medium. For the viability assay, 5 × 10^3^ HC and primary OA chondrocytes (pCH-OA) were seeded on white 96-well plates and either used as a control or treated with acetylshikonin, shikonin, or cyclopropylshikonin at different concentrations, ranging from 0.1 to 25 µM, for 24 h. For the proteome profiler assay and the protein and RNA isolation, 5 × 10^3^ HC and pCH-OA were seeded in 6-well plates and treated with 0.5 µM and 1.5 µM for 24 h, respectively. Only for the analyses of the phosphorylation status of MAPKs was the protein already isolated after 60 min.

### 3.4. Viability Assays

The dose-response curves were determined using the CellTiter 96 AQ_ueous_ Luminescence Assay (Promega, Madison, MA, USA). The primary OA chondrocytes (pCH-OA) and 5 × 10^3^ HC were seeded on white 96-well plates and either used as a control or treated with acetylshikonin, shikonin, or cyclopropylshikonin in various concentrations between 0.1 and 25 µM. After a 24 h incubation period, the viability assay was performed in accordance with the manufacturer’s protocol. Background reference values were derived from the culture media. Absorbance values were measured with the Lumistar microplate luminometer (BMG Labtech, Ortenberg, Germany). IC_50_ values were calculated with SigmaPlot 14.0 (Systat Software Inc., San Jose, CA, USA), using the four-parameter logistic curve.

### 3.5. Proteome Profiler Assay

Using the membrane-based proteome profiler human cytokine array kit (R&D Systems, Minneapolis, MN, USA), the relative levels of 36 cytokines, chemokines, and acute phase proteins were determined from the cell culture supernatants of HC and pCH-OA. To smooth out the patient-specific differences, the supernatants from three different primary cultures were pooled for analysis. The untreated controls were compared with 1.5 µM shikonin-treated chondrocytes.

### 3.6. Western Blot Analysis

After treatment with 0.5 µM and 1.5 µM, shikonin and its derivatives for 60 min, respectively, for 24 h, the whole cell protein extracts were prepared with lysis buffer (50 mM Tris-HCl pH 7.4, 150 mM NaCl, 1 mM NaF, 1 mM EDTA, 1% NP-40, 1 mM Na3 VO4) and protease inhibitor cocktail (P8340; Sigma Aldrich), subjected to SDS-PAGE, and blotted onto Amersham™ Protran™ Premium 0.45 µM nitrocellulose membrane (GE healthcare Life science, Little Chalfont, UK). Protein concentration was determined with the Pierce BCA Protein Assay Kit (Thermo Fisher Scientific) according to the manufacturer’s protocol. Primary antibodies against phospho-AKT^Ser473^, AKT, phospho-STAT3^Tyr705^, STAT3, phospho-ERK^Thr202/Tyr204^, ERK, phospho-JNK^Thr183/Tyr185^, JNK, phospho-p38^Thr180/Tyr182^, p38, survivin, Noxa, and cleaved-PARP (Cell Signaling Technology) were used. β-Actin was purchased from Santa Cruz (Santa Cruz Biotechnology, Santa Cruz, CA, USA). The blots were developed using a horseradish peroxidase-conjugated secondary antibody (Dako, Jena, Germany) at room temperature for 1 h and the Amersham™ ECL™ prime Western blotting detection reagent (GE Healthcare), in accordance with the manufacturers protocol. Chemiluminescence signals were detected with the ChemiDocTouch Imaging System (BioRad Laboratories Inc., Hercules, CA, USA), and images were processed with the ImageLab 5.2 Software (BioRad Laboratories Inc.). All full-length blots are presented in Appendix A.

### 3.7. Reverse Transcription Polymerase Chain Reaction (RT-PCR)

Total RNA was isolated 24 h after treatment with 1.5 µM shikonin and its derivatives using the RNeasy Mini Kit and DNase-I treatment according to the manufacturer’s manual (Qiagen, Hilden, Germany). Two µg RNA were reverse transcribed with the iScript-cDNA Synthesis Kit (BioRad Laboratories Inc., Hercules, CA, USA) using a blend of oligo(dT) and hexamer random primers. Amplification was performed with the SsoAdvanced Universal SYBR Green Supermix (Bio-Rad) using technical triplicates and measured by the CFX96 Touch (BioRad Laboratories Inc.). The following QuantiTect primer assays (Qiagen) were used for real time RT-PCR: MMP1, MMP3, IL6, cMyc, survivin, SOCS3, Sox9, cyclin D1, MEF2c, RUNX2, and c/EBPβ. The results were analyzed using the CFX manager software for CFX Real-Time PCR Instruments (Bio-Rad Laboratories Inc., version 3.1) software, and the quantification cycle values (Ct) were exported for statistical analysis. The results with Ct values greater than 32 were excluded from the analysis. Relative quantification of the expression levels was obtained by the ∆∆Ct method, based on the geometric mean of the internal controls ribosomal protein, large, P0 (RPL), and TATA box binding protein (TBP), respectively. The expression level (C_T_) of the target gene was normalized to the reference genes (ΔC_t_), and the ΔC_t_ of the test sample was normalized to the ΔC_t_ of the control (ΔΔC_t_). Finally, the expression ratio was calculated with the 2^−ΔΔCt^ method.

### 3.8. Statistical Analysis

The Student’s unpaired *t*-test and the exact Wilcoxon test were used to evaluate the differences between groups with the PASW statistics 18 software (IBM Corporation, Somers, NY, USA). Two-sided *p*-values (*p* < 0.001 ***; *p* < 0.01 **; *p* < 0.05 *) were considered statistically significant.

## 4. Conclusions

In summary, our results demonstrate for the first time that shikonin and its derivatives acetylshikonin and cyclopropylshikonin have extensive effects on inflammatory processes, MAPKs, and IL6/STAT3 downstream regulation in human healthy chondrocytes and primary OA chondrocytes. Our results provide important insights into the effects of shikonin derivatives in the context of OA-chondrocyte regulation and open new therapeutic perspectives with evidence that the pharmacological inhibition of the MAPK or IL6/STAT3 signaling pathways could reduce OA progression.

## Figures and Tables

**Figure 1 ijms-23-03396-f001:**
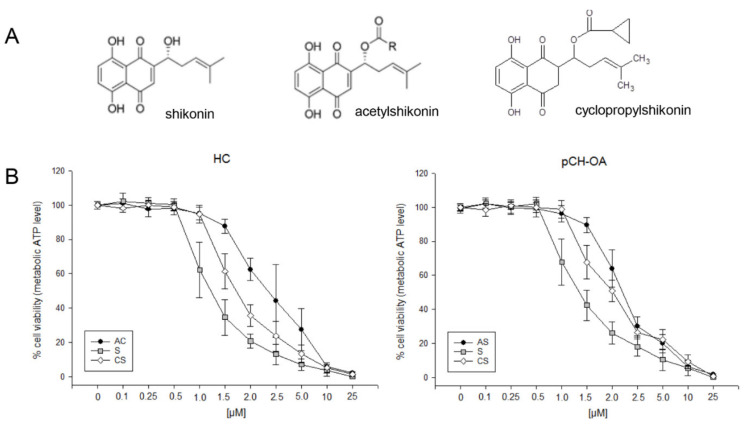
Influence of shikonin derivatives in chondrocyte viability. (**A**) Chemical structures of shikonin (S), acetylshikonin (AS), and cyclopropylshikonin (CS); (**B**) cell growth of healthy chondrocytes (HC) and human primary osteoarthritis chondrocytes (pCH-OA) was inhibited in a dose-dependent manner by shikonin derivatives (mean ± SD, *n* = 6, measured in biological quadruplicates).

**Figure 2 ijms-23-03396-f002:**
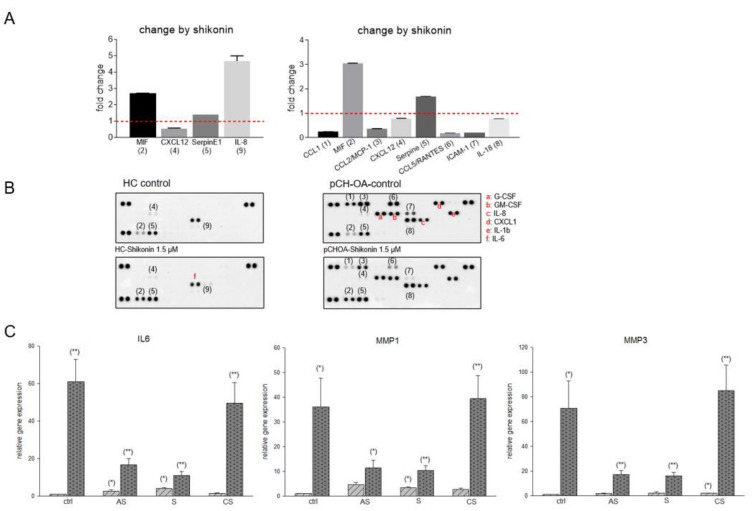
Modulation of IL1β-induced inflammatory cytokines and the expression of MMP1 and MMP3 in healthy and osteoarthritis chondrocytes. (**A**) Proteome profiler assay of healthy chondrocytes (HC) and human primary osteoarthritis chondrocytes (pCH-OA) after treatment with 1.5 µM shikonin revealed increased regulation of the IL-1β-induced inflammatory parameters. The supernatants of three different primary cultures were pooled for analysis. The numbers in parentheses mark relevant targets with their designation given in the bar chart (**B**). Expressed cytokines without relevant changes by shikonin but with existing differences between healthy and OA cells are marked (letters in red). (**B**) Standardized to the respective untreated controls, shikonin-induced fold changes are given. (**C**) Relative gene expression of IL6, MMP1, and MMP3 after treatment with 1.5 µM acetylshikonin (AS), shikonin (S), and cyclopropylshikonin (CS) for 24 h of HC (light grey striped) and pCH-OA (dark grey dotted) (mean ± SD, *n* = 6, measured in triplicates). Cell passages 1 and 2 of the human primary chondrocytes were used for experiments. Statistical significances are defined as follows: * *p* < 0.05; ** *p* < 0.01.

**Figure 3 ijms-23-03396-f003:**
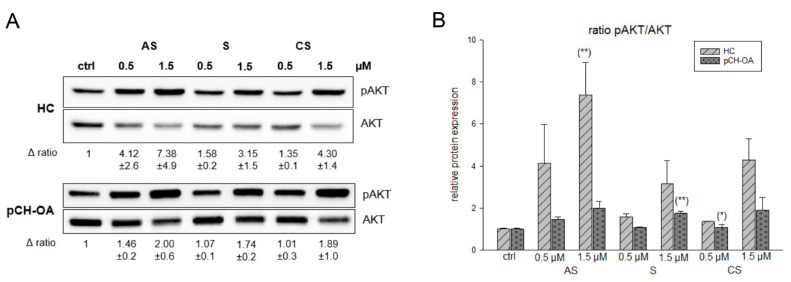
Shikonin derivatives induced an activation of AKT phosphorylation. (**A**) The AKT protein phosphorylation was evaluated by immunoblotting under untreated control conditions (ctrl) and in the presence of 0.5 µM and 1.5 µM acetylshikonin (AS), shikonin (S), and cyclopropylshikonin (CS) for 1 h in HC and IL-1β-stimulated pCH-OA cells. AKT was used as loading control. Δ ratio represents the fold change of pAKT/AKT normalized to controls (mean ± SD, *n* = 3). (**B**) shows the densiometric evaluation of all experiments in HC (light grey striped) and pCH-OA (dark grey dotted) cells. Statistical significances are defined as follows: * *p* < 0.05; ** *p* < 0.01.

**Figure 4 ijms-23-03396-f004:**
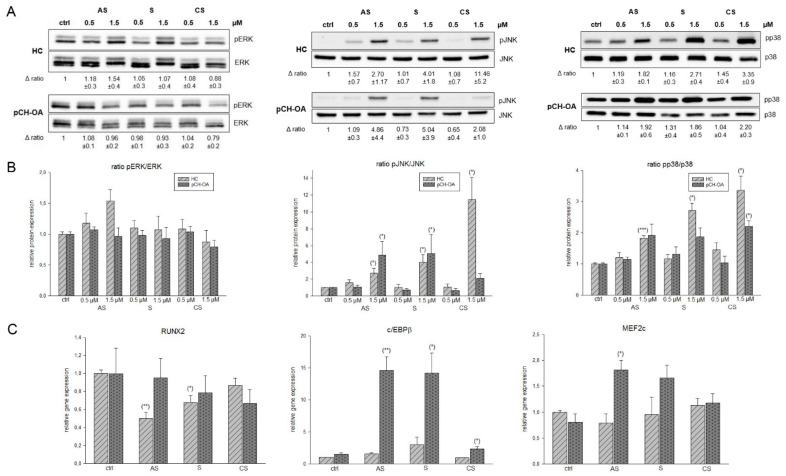
Shikonin derivatives induced an activation of MAPKs and their corresponding downstream targets. (**A**) The ERK/JNK/p38 protein phosphorylation was evaluated by immunoblotting under untreated control conditions (ctrl) and in the presence of 0.5 µM and 1.5 µM acetylshikonin (AS), shikonin (S), and cyclopropylshikonin (CS) for 1 h in HC and IL-1β-stimulated pCH-OA cells. The unphosphorylated ERK/JNK/p38 were used as loading controls. Δ ratio represents the fold change of phosphorylated and non phosphorylated kinases normalized to controls (mean ± SD, *n* = 3). (**B**) shows the densiometric evaluation of all experiments in HC (light grey striped) and pCH-OA (dark grey dotted) cells. (**C**) Relative gene expression of the downstream targets RUNX2, c/EBPβ, and Mef2c after treatment with shikonin derivatives for 24 h (mean ± SD, *n* = 6, measured in triplicates). Statistical significances are defined as follows: * *p* < 0.05; ** *p* < 0.01; *** *p* < 0.001.

**Figure 5 ijms-23-03396-f005:**
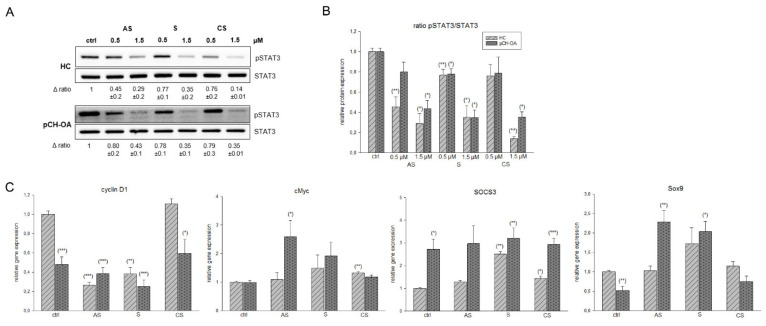
Inhibition of STAT3 phosphorylation and the influence of shikonin derivatives on STAT3 downstream targets. (**A**) The STAT3 protein phosphorylation was evaluated by immunoblotting under untreated control conditions (ctrl) and in the presence of 0.5 µM and 1.5 µM acetylshikonin (AS), shikonin (S), and cyclopropylshikonin (CS) for 1 h in HC and IL-1β-stimulated pCH-OA cells. STAT3 were used as loading controls. Δ, fold change normalized to controls (mean ± SD, *n* = 3). (**B**) shows the densiometric evaluation of all experiments in HC (light grey striped) and pCH-OA (dark grey dotted) cells. (**C**) Relative gene expression of the downstream targets cyclin D1, cMyc, SOCS3, and Sox9 after treatment with shikonin derivatives for 24 h (mean ± SD, *n* = 6, measured in triplicates). Statistical significances are defined as follows: * *p* < 0.05; ** *p* < 0.01; *** *p* < 0.001.

**Figure 6 ijms-23-03396-f006:**
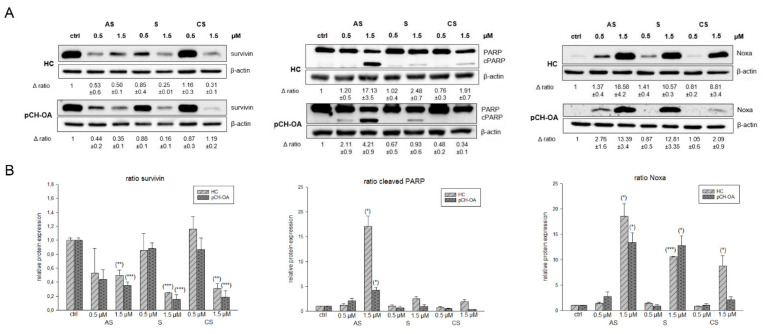
Influence on apoptotic targets. (**A**) Relative protein expression analysis of survivin, Noxa, and cleaved PARP after treatment with 0.5 µM and 1.5 µM acetylshikonin (AS), shikonin (S), and cyclopropylshikonin (CS) for 24 h in HC and IL1β-stimulated pCH-OA cells. Untreated control cells (ctrl) served as reference value and β-actin as loading control (mean ± SD, *n* = 3). (**B**) Shows the densiometric evaluation of all experiments in HC (light grey striped) and pCH-OA (dark grey dotted) cells. Statistical significances are defined as follows: * *p* < 0.05; ** *p* < 0.01; *** *p* < 0.001.

## Data Availability

Not applicable.

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
