# Peer review of "Shikonin Derivatives Inhibit Inflammation Processes and Modulate MAPK Signaling in Human Healthy and Osteoarthritis Chondrocytes"

_ijms, 2022, doi:10.3390/ijms23063396_

Round 1
Reviewer 1 Report
The article Manuscript ID: ijms-1607961 entitled "Shikonin derivatives inhibit inflammation processes and modulate MAPK signaling in human primary OA chondrocytes" can be accepted for publication in the “International Journal of Molecular Sciences” after major revisions.
The abstract is not complete. The part concerning methods should be described. Please include.
Introduction:
The first part of the introduction concerning the definition of OA is superficial and not supported by the correct and updated references, please improve it. The pathogenesis of OA should be re-written including other and more up-date studies. See for instance:
Hunter DJ, Bierma-Zeinstra S. Osteoarthritis. Lancet. 2019;393(10182):1745-1759. doi: 10.1016/S0140-6736(19)30417-9.
Chow YY, Chin KY. The Role of Inflammation in the Pathogenesis of Osteoarthritis. Mediators Inflamm. 2020 Mar 3;2020:8293921. doi: 10.1155/2020/8293921.
Hu Q, Ecker M. Overview of MMP-13 as a Promising Target for the Treatment of Osteoarthritis. Int J Mol Sci. 2021 Feb 9;22(4):1742. doi: 10.3390/ijms22041742.
There is no mention about the therapeutic options for the management of OA
The aim of the study should be improved
Materials and methods:
The Authors should include a paragraph explaining the treatment performed in the study, in a specific manner, the timing of treatment, and how and why the concentrations of the compound are selected…
Results:
Figure 2A is not understandable by the information provided from the authors..
Please re-check the figure legends, in some cases they don’t provide enough information to be clear for the reader.
Discussion:
This section could be improved, mainly discussed, and enriched with results derived from other studies.
This part should consider what all of the findings taken together mean. What are the larger implications and, importantly, what are the limitations.
The authors need to clarify what advance or advances in knowledge were made by this work.
Author Response
We would like to thank the reviewers for their efforts and valuable suggestions. We hope that we have interpreted the comments correctly and that the modifications of the manuscript will meet the reviewers’ expectations and are sufficient for publication.
I refer here to the word file, where we have dealt with the individual comments point by point.

Reviewer 2 Report
Shikonin and its derivatives affect the viability of healthy and osteoarthritic chondrocytes when applied in concentrations higher than 0.5 µM. Yet, in the presented experiments the mainly used concentration was 1.5µM, a concentration which reduces cell viability by more than 50 %. The reasons should discussed.
Shikonin reduces proinflammatory proteins in affected chondrocytes. But some, eg IL18 also are upregulated. How could this be explained?
Shikonin treatment activated the MAPK pathway and increased the phosphorylation AKT, Erk and p38 in osteoarthric chondrocytes. MMP3 expression – a protein which degrades cartilage - went up. Apoptotic proteins are upregulated and the antiapoptotic molecule suvivin is downregulated in ostearthritic chondrocytes. These results have to be discussed.
The effects induced by shikonin diverge between healthy and osteoarthritic chondrocytes. What might be the reasons?
The work is based on a lot of data. For correct understanding their impact on intracellular regulation , they should be discussed in detail.
For a correct classification of all recorded shikonin induced effects in osteoarthritic chondrocytes vs healthy chondrocytes, a comparative graphical abstract would be helpful. This possibly could emphasise the impact of shikonin as a potential drug.
Author Response

(The authors gave the same response as above.)

Reviewer 3 Report
The authors showed that in chondrocytes multiple IL-1β-induced pro-inflammatory pathways can be inhibited by shikonin. This was seen in healthy control and in OA chondrocytes, but the shikonin effects were strongest in healthy control chondrocytes.
My comments:
1) The authors should be careful with their claims regarding the role of inflammation in osteoarthritis (OA) and the relation between their findings and OA pathology. They only studied IL-1β-induced inflammatory pathways in chondrocytes. Joint inflammation (low-grade) was found in a large % of OA patients, but treatment of OA patients with specific IL-1 inhibitors or other inhibitors of inflammation did not stop OA progression. Here are examples of false claims:
- a) The title suggests that shikonin has specific effects in OA chondrocytes, while the results show in most cases the same responses, although somewhat lower, in OA chondrocytes, as compared to the healthy controls. Therefore the word “OA” should be removed from the title, or both Healthy and OA chondrocytes should be mentioned.
- b) The first 3 lines of the abstract starts with OA “is the most common joint disorders”(typo!) and “few effective therapeutic approaches” and then: “therefore, we investigated the effect of shikonin…. I don’t see the link between OA and the experiments described in this study. It is still important to get more insight in IL-1-induced inflammatory pathways in chondrocytes and look for OA-specific differences in their regulation.
- c) lines 188-189: “The IL6/signal transducer and activator of transcription 3 (STAT3) signaling is one of the key pathways in OA pathogenesis.” I do not think that this statement is generally accepted knowledge. Moreover, references are lacking.
d)In the conclusions (lines 341-347) “Our results bring important insights into OA pathogenesis and open new therapeutic perspectives with the evidence that pharmacological inhibition of MAPK or IL6/STAT3 pathways could reduce OA progression.” Which new insights in OA pathogenesis? Moreover, cEBPβ was increased with high significance by shikonin specifically in the OA chondrocytes, and this could lead to cartilage matrix degradation (lines 168-172). This differential regulation in OA chondrocytes is interesting, but did not get much attention in the manuscript.
2) Only in one single sentence, in the Materials and Methods part of this manuscript, it is mentioned that the chondrocytes in monolayer were stimulated with 10 ng/ml IL-1β one hour before shikonin treatments. To my opinion this should be mentioned in the legends of every figure, because the figures should be self-explaining. So please make this clear as follows:
In the legends of figure 2 in line 130: Inhibition of IL-1β-induced inflammatory cytokines and…..
In the legends of figure 3 in line 147: under control (ctrl) conditions (IL-1β alone) and in the presence of……..
In the legends of figure 4 in line 180: under control (ctrl) conditions (IL-1β alone) and in the presence of….
In the legends of figure 5 in line217: Inhibition of IL-1β-induced STAT3 phosphorylation……
In the legends of figure 6 in line 241: Control cells (ctrl) stimulated with IL-1β alone served as reference value and β-actin (look for typo here!)…..
It is not clear to me whether the IL-1β-background is used in shikonin dose-response studies in figure 1. This should be the case, because such a high dose of IL-1β itself influences viability, and maybe also the IC50 of shikonin.
3) In figures 2, 3,4,5 and 6, “HC (light grey striped) and pCH-OA (dark grey dotted)” is mentioned under A) (immune stainings), but this should be in B) (densitometric evaluation).
4) In the legends of figures 1, 2, 4 and 5 we can read “n = 6, measured in triplicates”, Does this mean that chondrocytes 6 different donors were used for both HC and pCH-OA? Or were all HC from the same donor? Or were there 6 different plates with cells of the same donor? Or were the cells obtained from first and second passage of cells from the same donor? Please make this clear. In Figures 3 and 6 we can only read “(n = 3)”. Does this mean that cells of three different donors were used, or did the authors show triplicates of cells from the same donor? Please make this clear. Last question about numbers of donors: in the legends of figure 2 we can read that n = 6, but in the Materials and Methods section (lines 293-299) we can read that in the proteome profiler assay to smooth out patient-specific differences, the supernatants of three different primary cultures were pooled for analysis. Does n = 6 mean that two pools were analysed, or six pools were analysed, or that one pool was analysed in duplicate? The smoothing out was done for the OA patients material, but how many different HC were used? OA patients were aged 73.4 ± 5.8 years. How many different HC were used (should be mentioned in line 271 in materials and methods). What was the age of the healthy HC?
5) In paragraph 3.2 of the Materials and Methods section, line 275 we can read that in the chondrocyte growth medium there was 0.01% TGF-β(10 ng/µl). If 10 ng/µl is the end-concentration this is a very high concentration of TGF-β. Did the authors mean to write 10 ng/ml? or do they mean that a 0.01% dilution of a 10 ng/µl stock (= 1 ng/ml end-concentration) was made? Also make this clear for the FGF concentration mentioned in the same line.
Line 277: At what density were the cells seeded for the experiments? And, in what medium was used during the stimulations with IL-1 and shikonin, was this in the chondrocyte growth medium with all its additions, or in a more basic medium?
6) I would expect some discussion on the differences in the functionality of the three different shikonin derivatives.
Author Response

(The authors gave the same response as above.)

Reviewer 4 Report
The authors of this study have investigated the effects of shikonin derviatives on healthy and osteoarthritic chondrocytes- The data show that the derivative are inhibit IL-6 production and MMP-1 and -3 activity, whilst downstream targets of IL6/STAT3 pathway are inhibited, particularly pSTAT3, and leads to upregulation in SOX9 expression. This a potential pharmacological mechanism for inhibiting the progression of osteoarthritis.
This is good introductory publication for understanding how shikonin and its derivatives can reduce osteoarthritis progression. The authors need to answer the following points
- What was the rationale for using 10ng/mL IL-1β? Should this concentration not be reduced, as the point to apply these treatments would be at the earliest stages of the disease to have a greater effectiveness.
- What was the rationale to use monolayer chondrocytes rather than 3D culture of the chondrocytes to understand the effect of shikonin derivatives ? Normally, chondrocytes dedifferentiate with time in culture and requires redifferentiation to restore phenotype. Can you present preliminary data to show that there is no differences between monolayer and pellet culture of chondrocytes.
- Figure 1, please show normal healthy and osteoarthritic chondrocyte growth in the figure to visualise the difference between normal and shikonin derivative treatment.
- For gene expression data in Figure 2c, what is the data significant to ? Is this significant between the control and treated group or between the two chondrocyte types for the specific treatment ? Please use lines between groups to clearly show significant groups.
- It is unclear about the significance of the apoptosis data in the general scheme of the previous presented. It is known that PI3K/AKT pathway controls apoptotic genes in osteoarthritis. Thus, can the authors rephrase section 2.5 with a clear discussion of the data that relates to the pathways investigated ?
- An extra figure showing the pathways under the control of shikonin derivatives and summarizing the data from this publication should be included. This would help readers to understand the significance of the results.
Author Response

(The authors gave the same response as above.)

Round 2
Reviewer 1 Report
The authors have modified and improved the manuscript according to the reviewer's suggestions.
Reviewer 4 Report
The authors have answered my review questions appropriately.